# Why Do We Not Wear Masks Anymore during the COVID-19 Wave? Vaccination Precludes the Adoption of Personal Non-Pharmaceutical Interventions: A Quantitative Study of Taiwanese Residents

**DOI:** 10.3390/medicina60020301

**Published:** 2024-02-09

**Authors:** Lee-Xieng Yang, Chia-Yuan Lin, Wan-Zhen Zhan, Bo-An Chiang, En-Chi Chang

**Affiliations:** 1Department of Psychology, National Chengchi University, Taipei 11605, Taiwan; janet2236388@gmail.com (W.-Z.Z.); pachiang@nccu.edu.tw (B.-A.C.); 109702038@nccu.edu.tw (E.-C.C.); 2Research Center for Mind, Brain, and Learning, National Chengchi University, Taipei 11605, Taiwan; 3Department of Psychology, University of Huddersfield, Huddersfield HD1 3DH, UK; c.lin@hud.ac.uk; 4Centre of Cognition and Neuroscience, University of Huddersfield, Huddersfield HD1 3DH, UK

**Keywords:** COVID-19, risk perception, Peltzman effect

## Abstract

*Background and Objectives*: This study examined whether the decline in people’s adoption of personal NPIs (e.g., mask wearing) results from the preclusion by vaccination. This study also incorporates the concepts of risk perception and the risk-as-feelings model to elucidate the possible mechanisms behind this preclusion. *Materials and Methods*: Two cross-sectional surveys (N = 462 in Survey 1 and N = 505 in Survey 2) were administered before and during the first outbreak of COVID-19 in Taiwan. The survey items were designed to measure participants’ perceived severity of COVID-19, worry about COVID-19, intention to adopt personal NPIs, and attitudes toward COVID-19 vaccines. Utilizing the risk perception framework, we conducted multigroup SEM (Structural Equation Modeling) to construct the optimal structural model for both samples. *Results and Conclusions*: The multigroup SEM results showed that worry (i.e., the emotional component of risk perception) fully mediates the influence of the perceived severity of COVID-19 (i.e., the cognitive component of risk perception) on the intention to adopt NPIs in both surveys [*z* = 4.03, *p* < 0.001 for Survey 1 and *z* = 2.49, *p* < 0.050 for Survey 2]. Before the outbreak (i.e., Survey 1), people’s attitudes toward COVID-19 vaccines showed no significant association with their worry about COVID-19 [*z* = 0.66, *p* = 0.508]. However, in Survey 2, following the real outbreak of COVID-19, people’s attitudes toward COVID-19 vaccines negatively predicts their worry about COVID-19 [*z* = −4.31, *p* < 0.001], indirectly resulting in a negative effect on their intention to adopt personal NPIs. This suggests the occurrence of the Peltzman effect. That is, vaccination fosters a sense of safety, subsequently diminishing alertness to COVID-19, and thus reducing the intention to adopt personal NPIs.

## 1. Introduction

Undoubtedly, COVID-19 has caused significant harm to public health, yet it also prompts a critical examination of the general public’s compliance to preventive measures, particularly nonpharmaceutical interventions (NPIs). NPIs encompass personal measures, physical distancing, movement restrictions, and specific protective measures for vulnerable groups [1], which serve as essential safeguards against direct contact with COVID-19 viruses via physically separating individuals and viruses. Before COVID-19 vaccines were available publicly, NPIs stood as the sole effective measures to contain the pandemic [2].

While previous research suggested that optimal protection comes from the combined implementation of both NPIs and vaccination [3], and NPIs could prove effective against various respiratory tract infections [4], NPIs seemed to be less favored. This might be true for certain NPIs, such as lockdowns, which have significant socioeconomic costs and may adversely affect individual well-being [5,6]. However, personal NPIs, such as mask-wearing, also appeared to be less favored [7] despite their cost-effectiveness in preventing COVID-19 transmission [8]. Previous studies showed that physical discomfort and doubts about effectiveness were reasons for people’s reluctance to wear masks [9]. Additionally, cultural factors, such as being unfamiliar with mask-wearing in Western cultures, could contribute to hesitancy in adopting personal NPIs, as shown in the UK [10]. In Germany, people’s willingness to wear masks could be increased if masks were legally required [11]. Surprisingly, in Taiwan, where wearing masks in public was common, a previous study revealed that 48% of over 500 participants exhibited a high motivation to receive vaccinations but a low motivation to adopt personal NPIs [12].

Apart from the above reasons, a further reason for people to be even less willing to comply with personal NPIs could be that they believed that they have acquired immunity to SARS-CoV-2 (i.e., the COVID-19 virus) via vaccinations. This is apparently not the best practice, as new variants of the virus emerge continuously [13], not to mention the potential for immune evasion, wherein our immune system may forget a known SARS-CoV-2 variant [14]. And, as indicated in the aforementioned studies [3], the best practice should be a combined implementation of both vaccination and NPIs, as this practice can most effectively reduce the transmission of COVID-19.

If people’s unwillingness to adopt personal NPIs is because they think they have acquired immunity, then this may highlight a cognitive gap between public common sense and scientific facts, which potentially exposes the public to hazards. This resonates with the Peltzman effect [15], suggesting that individuals are more likely to engage in risky behaviors when security measures are mandated. For instance, when seatbelt use is mandated, people may drive with less attentiveness. Thus, it is plausible that the decline in personal NPIs results from the “safe feeling” brought by COVID-19 vaccinations. However, previous studies rarely examined the relationships between vaccination and personal NPIs. Therefore, the purpose of this study is to examine whether the Peltzman effect occurred during COVID-19 outbreaks. Additionally, we aim to elucidate how the Peltzman effect may manifest via risk perception and the risk-as-feelings model [16].

### 1.1. Risk Perception and Risk-As-Feelings

Risk perception refers to an individual’s feeling and understanding of objective risks in the outside world [17]. While the mental constructs of the risk perception of COVID-19 may vary under different research concerns, it is commonly acknowledged that it consists of cognitive and emotional components [18]. For example, the cognitive component can refer to the perceived severity, fatality rate, and transmission rate of COVID-19, while the emotional component can refer to the fear and worry over being infected. These two components in COVID-19 risk perception have been reported to be positively correlated with each other [18].

Presumably, people’s risk perception of COVID-19 should inform their reactions to the pandemic. Previous research has shown that an individual’s willingness to receive the COVID-19 vaccination was positively correlated with their emotional responses, such as fear and worry [19,20,21]. Additionally, emotions related to COVID-19, including fear, anxiety, and worry, have been reported to predict an individual’s willingness to adopt NPIs [22,23,24,25]. These results collectively imply that individuals’ reactions (i.e., receiving vaccination or adopting NPIs) to COVID-19 may be led by the emotional component of their risk perception. This is consistent with the predictions of the risk-as-feelings model [16]. That is, when a risk is unknown and uncommon, our reaction to it is instinctive and intuitive [26]. However, when people get used to a risk, such as prolonged exposure, their perceived risk decreases [27]. According to the risk-as-feelings model, one situation for the occurrence of the Peltzman effect could be that people become less worried about being infected due to having received vaccinations, hence decreasing their intention to adopt personal NPIs. In this study, we aim to verify this hypothesis by conducting SEM (Structural Equation Modeling) on the collected data. 

### 1.2. Taiwan As an Ideal Setting to Investigate the Peltzman Effect on Personal NPIs

The Taiwanese government implemented stringent NPIs, including border quarantine, from early 2020 to early 2022. As a result, the curve of daily confirmed cases and death cases remained flat (only around 60 new cases per day) [28]. Concurrently, the government achieved a vaccination coverage rate exceeding 80% of the population with at least one dose of the COVID-19 vaccine by the end of January 2022 (see the website of Our World in Data; https://ourworldindata.org/grapher/share-people-vaccinated-covid?country=~TWN, (accessed on 1 February 2024). Around the end of January 2022, Taiwan shifted its prevention policy from zero-COVID-19 to coexisting with the virus. Although compulsory NPIs (e.g., border quarantine, school closure, banning public gathering, etc.) were lifted, personal/voluntary NPIs, including mask wearing, social distancing, and hand hygiene, remained in effect.

The first significant outbreak of COVID-19 in Taiwan only occurred in May 2022, caused by the Omicron variant, resulting a sudden surge of 60,000 daily cases [16]. Before this outbreak, most Taiwanese people had not been infected and had already received the protection of COVID-19 vaccines. This makes Taiwan as an ideal setting to explore the occurrence of the Peltzman effect on the adoption of personal NPIs in relation to vaccination. 

In this circumstance, if there is no difference in people’s intention adopt personal NPIs before and during the outbreak, the Peltzman effect may not be present. Conversely, if Taiwanese people decline to adopt personal NPIs during the outbreak, it will support the occurrence of the Peltzman effect. Most Western countries are less suitable for investigating the Peltzman effect, given the unsatisfactory vaccination rate in early COVID-19 outbreaks [3]. In contrast, Taiwan is an ideal choice, as most Taiwanese have received vaccination and were not infected before the initial COVID-19 outbreak. It is important to note that while Taiwan provides a fitting setup to examine the Peltzman effect, the aim is not to suggest exclusivity but rather to elucidate the reasons why Taiwanese samples are particularly suitable for this study.

In early 2022, we conducted a cross-sectional survey to measure people’s risk perception of COVID-19, their attitudes toward COVID-19 vaccines, and their intention to adopt personal NPIs. Unexpectedly, the Omicron outbreak occurred a couple of months after our first survey. Thus, we took advantage of this great opportunity to conduct a second cross-sectional survey. Just like a natural experiment (with the independent variable as the timing of survey before and during the outbreak of Omicron), we can compare the results of these two surveys to see how the outbreak of Omicron influenced people’s risk perception and their reactions to COVID-19. Specifically, the occurrence of the Peltzman effect is examined.

## 2. Materials and Methods

### 2.1. Study Sample

For Survey 1, data were collected by three of our authors (Zhan, Chiang, and Chang) who shared the URL for access to the online questionnaire on the three most popular social media platforms in Taiwan: Facebook, PTT, and Dcard. The data collection period was from 22 January to 28 February 2022, during which three prize draws were conducted. Participants had the chance to win one of ten TWD 300 (≅USD 10), ten TWD 200 (≅USD 6.67), and five TWD 200 lottery prizes. To maximize the response rate upon questionnaire release, participants who did not win in one prize draw were automatically entered into the next prize draw. In total, 472 participants were recruited for the first survey. Participants were precluded from data analysis if they had at least one missing value. As a result, 462 participants were included in data analysis.

Survey 2 was conducted from 18 May to 29 June 2022, using the same prize draw procedure as in Survey 1. A total of 517 participants were recruited for the second survey. With the same criterion used in Survey 1, 505 participants were included in the data analysis. All participants were adults (>18 years old) and gave written consent before completing the survey. Data collection was anonymous, and no personal information was collected except age. A priori power analysis for SEM showed that our sample sizes were sufficiently large (for the expected power set as 0.80 and α = 0.05) for both surveys. Details of the power analysis will be provided in the Results section. As the outbreak of Omicron occurred in May 2022, participants in Survey 1 could represent the population in Taiwan who had never experienced the outbreak of COVID-19, whereas those in Survey 2 could represent the population who encountered the outbreak of COVID-19 for the first time. 

### 2.2. Study Design

In this study, we administered two cross-sectional surveys with a same questionnaire to investigate participants’ perceived severity of COVID-19, perceived worry of COVID-19, intention to adopt personal NPIs, and attitudes toward COVID-19 vaccines. 

### 2.3. Instruments

The questionnaire used in this study is listed in Table A1 in Appendix A, which consists of two parts. Part A aims to collect demographic data, including age and the number of COVID-19 vaccine doses taken. In Survey 2, as the daily confirmed case increased significantly, an additional question was included to inquire about how many times they had tested positive for COVID-19.

According to the concept of risk perception and the risk-as-feelings model, the Part B items (from Q1 to Q20) in this questionnaire were designed to measure four mental constructs. The first was the cognitive component of the risk perception of COVID-19, which was measured by items Q7–Q9 and coded as Severity in Table A1 and in the SEM results. The second was the emotional component of the risk perception of COVID-19, measured by items Q2–Q4 and Q19 and coded as Worry. The third was the intention to adopt personal NPIs, measured by items Q16–Q18 and coded as NPI. The last was the attitudes toward COVID-19 vaccines, measured by items Q10–Q12 and coded as Vaccine (Q13 was precluded from Vaccine for seeking the optimal model fit). We conducted multigroup SEM for our data. The descriptive statistics of these items and the reliability for measuring each construct (Cronbach’s α) are listed in Table 1. The reliabilities are mostly ≥0.70, except for Severity in Survey 2. This might be because of the high skewness of the distributions for Q7–Q9 (see Figure 1). In the Results section, we will report the reliability and validity estimated in the framework of SEM. The detailed description of each item is shown in Table A1 in Appendix A.

### 2.4. Statistical Analyses

First, comparisons of the general characteristics of these two samples were described. Second, we used the R package {lavaan} (ver. 4.3.1) [29] to conduct multigroup SEM for the data from these two samples. We tested to what extent these two samples could be viewed as equal with different models, assuming that the structural models for these two samples have the same configuration and assuming that the parameters have the same values for these two samples. This check can help us explain the differences between the two samples.

## 3. Results

### 3.1. Comparisons of the Demographic Characteristics between the Two Samples

The age distributions in Survey 1 (M = 28.67 years, SD = 7.17 years) and Survey 2 (M = 28.41 years, SD = 7.79 years) were similar, and the difference between their mean ages was not significant [$t_{\left(908\right)}=0.51$, p=0.61]. More participants received three doses of COVID-19 vaccines in Survey 2 compared to Survey 1 (see Table 1 for more details). The proportion of participants testing positive in Survey 2 was 6.93%, which was close to the COVID-19 prevalence rate of 7.99% in Taiwan at that time [27].

### 3.2. Multigroup SEM

We conducted multigroup SEM to assess to what extent these two samples exhibit invariance by comparing four structural models. These models assumed configural invariance for the two samples (i.e., only the configuration of the structural model is the same for both of them), measurement invariance (i.e., the configuration of the structural model and the factor loadings on the observed items were the same), scalar invariance (i.e., the factor loadings and the intercepts of the latent variables were the same), and residual invariance (i.e., the factor loadings, the intercepts of latent variables, and the residual of each regression formula were the same). Apparently, the latter models set more constraints on invariance than the previous ones do. Thus, we compared each pair of models sequentially, from the model with the least constraint to the one with the most constraint. 

The results showed that the modeling performance started to significantly drop in the model assuming scalar invariance, compared with the model assuming measurement invariance [ΔAIC = 219 and ΔBIC = 175, $\chi_{9}^{2}=237.17$, p<0.01]. It was suggested that measurement invariance was held for these two samples. That is, these two samples were equivalent in terms of the constitution of the mental constructs of our interest, even though the data were collected in a cross-sectional design. This result also provided support for the construct validity of our questionnaire. In the model assuming measurement invariance, all parameters except the factor loadings on the observed items were estimated separately for each sample. 

#### 3.2.1. Measurement Model and Structural Model

The measurement model for the two samples consisted of four latent variables. They were worry about COVID-19 (indicated by items Q2, Q3, Q4, and Q19 in the questionnaire), severity of COVID-19 (indicated by Q7, Q8, and Q9), attitudes toward COVID-19 vaccines (indicated by Q10, Q11, and Q12 (Q13 was not included because the loading of it for constructing the latent variable of attitudes toward vaccines was not significant), and the intention to adopt personal NPIs (indicated by Q16, Q17, and Q18). All item scores were transformed into z-scores before constructing the latent variables. The composite reliabilities of the four latent variables in the measurement model for Survey 1 were mostly satisfactory (i.e., ≥0.70, according to [30]) [0.82, 0.66, 0.81, and 0.81, respectively, for the latent variables Worry, Severity, NPIs, and Vaccines]. The measurement model for Survey 2 also demonstrated good reliabilities for the four latent variables [0.81, 0.53, 0.83, and 0.81, respectively, for Worry, Severity, NPIs, and Vaccines]. The lower reliability for Severity might be due to the high skewness of the distributions for Q7, Q8, and Q9 (see Figure 1). The reliability for Severity can be raised to 0.76 (in Survey 1) and 0.70 (in Survey 2) by using the square roots of the scores of Q7, Q8 and Q9, suggesting that the original low reliability is due to the high skewness of the score distribution. However, in order to maintain consistency across all items, we continued to use z-scores.

Both measurement models exhibit a good construct validity, as the average variance extracted for Worry, Severity, NPIs, and Vaccines are all >0.5 [31], with 0.56, 0.63, 0.58, and 0.61 for Worry, Severity, NPIs, and Vaccines in Survey 1, and 0.53, 0.53, 0.61, and 0.58 for the same latent variables in Survey 2. See Table 2 for the statistics of the factor loadings, all of which were significant.

The optimal structural model performed well, as shown by the goodness-of-fit indices [$\chi_{127}^{2}=276.11$, p<0.01, CFI=0.97, TLI=0.96, RMSEA=0.05, and SRMR=0.05]. The scheme of this model can be seen in Figure 2. This structural model consists of 4 latent variables and 13 observed items in total. A priori power analysis for this model, with the anticipated power set as 0.80, effect size set as 0.30, and type I error set as 0.05, suggested that the minimum sample size should be 166. Both of our surveys recruited more than 166 participants, guaranteeing the power of this structural model to be at least 0.80.

#### 3.2.2. Worry Fully Mediated the Influence of Severity on NPIs

The two latent variables, Severity and Worry, were treated as the cognitive and emotional components of the risk perception of COVID-19. In Figure 2, no matter which sample, Severity positively predicts Worry. This is consistent with the previous finding [18]. NPIs, representing the intention to adopt personal NPIs, is predicted by Worry directly, but not by Severity directly, suggesting that the reaction to COVID-19 is driven by the emotional component of the risk perception of COVID-19. This finding supports the risk-as-feelings model. However, two points deserve attention. First, NPIs was driven by Worry in Survey 2. This result suggests that the participants in Survey 2 had not yet been habituated to COVID-19. This was in line with the fact that the Omicron outbreak was the initial, severe outbreak of COVID-19 in Taiwan. Second, while there was no direct effect of Severity on NPIs, the indirect effect of Severity on NPIs was significant, with Worry as the mediator for both samples. Thus, it is not appropriate to argue that the cognitive component of risk perception has nothing to do with the reaction to a risk. Instead, the effect of the cognitive component was fully mediated by the emotional component, hence elaborating the theoretical relationships between the risk-as-analysis and risk-as-feelings models [16]. 

#### 3.2.3. Worry Increased during the Outbreak

In order to determine if there was any change in people’s perceived severity of COVID-19, worry about COVID-19, attitudes toward COVID-19 vaccines, and intention to adopt NPIs, we compared the means of the four latent variables between the two surveys. The results showed that, compared with Survey 1, the participants in Survey 2 perceived COVID-19 as less severe [z=−4.53, p<0.01], were more worried about COVID-19 [z=16.47, p<0.01], but would be less willing to adopt the personal NPIs [z=−6.67, p<0.01]. The attitudes toward COVID-19 vaccines were not changed between these surveys [z=−1.64, p=0.10]. The decline in the intention to adopt NPIs might not result from the habituation to COVID-19, as the worry level in Survey 2 was higher than that in Survey 1. If people had become used to COVID-19, the worry level should have been lower in the second survey. Also, Worry still predicted NPIs in Survey 2, as it did in Survey 1. These results converge toward rejection of the hypothesis that the increase in willingness to adopt personal NPIs results from people’s habituation to COVID-19.

#### 3.2.4. Vaccine Attitudes Affected NPI Adoption, Implying the Peltzman Effect

Now we turn to verify the hypothesis that the decline in the adoption of NPIs may reflect the Peltzman effect brought about by vaccines. As shown in Figure 2, there are two regression paths from Vaccines to Worry and NPIs. In Survey 1, the regression weight from Vaccines to Worry was not significant [z=0.66, p=0.508], suggesting that the participants’ worry about COVID-19 could not be predicted by their attitudes toward vaccines. However, the direct effect of Vaccines on NPIs was significant [z=3.46, p<0.001], suggesting that the more positive attitudes toward vaccines were associated with a greater willingness to adopt personal NPIs. This is not surprising, as vaccines and NPIs are both prevention measures for COVID-19. Those who believed in the effect of vaccines on containing COVID-19 would be more likely to adopt personal NPIs to minimize the contact with COVID-19.

In Survey 2, Vaccines negatively predicted Worry [z=−4.31, p<0.001], meaning that a more positive attitude toward vaccines was associated with less worry about COVID-19. Perhaps this was because people in Survey 2 realized the protective effect of vaccination (e.g., mild symptoms) during the outbreak. As Worry positively predicted the intention to adopt NPIs [z=6.01, p<0.001], a decreasing worry level led to a lower intention to adopt NPIs. This inference was supported by the significant negative indirect effect of Vaccines on NPIs with the mediation of Worry [z=−4.00, p<0.001]. This negative effect suggested the occurrence of the Peltzman effect. In this case, people felt safer after having received COVID-19 vaccination, which, in turn, reduced their alertness and decreased their intention to continue to adopt personal NPIs. One important point to note is that the direct effect of Vaccines on NPIs was positive and significant [z=2.38, p=0.017]. This indicated that the effect of Vaccines on NPIs was not entirely replaced by Worry. Thus, it is suggested that worry about COVID-19 served as a partial mediator in the path from the attitudes toward vaccines to the intention to adopt NPIs. The worry about COVID-19 acted as a full mediator in the path from the perceived severity of COVID-19 to the intention to adopt NPIs. 

## 4. Discussion

Despite the facts that various NPIs could all effectively reduce the number of people infected with COVID-19 [32]; that when more NPIs were implemented together, there was a greater reduction in the number of contagious people [33]; and that mask-wearing was also effective against multiple respiratory tract infections, not only COVID-19 [4], people still declined to adopt personal NPIs during the COVID-19 waves, which motivated us to conduct this study. In addition to the aforementioned reasons, including physical discomfort and concerns about social norms, the results of this study revealed another reason, namely the Peltzman effect. The multigroup SEM results not only support the occurrence of this effect, but also provide a psychological mechanism to explain how it occurs. Several findings in our study align with previous studies. First, we replicated the positive correlation between the cognitive and emotional components of the risk perception of COVID-19 [18]. Second, we confirmed the prediction from the risk-as-feelings model by demonstrating that Worry can predict NPIs. Third, we found that the influence of Severity on NPI is fully mediated by Worry, suggesting that our cognition plays a role in shaping our reactions to a hazard, but its impact is fully mediated by emotion. Our results can contribute to the refinement of the risk-as-feelings model.

Fourth, and perhaps most importantly, we found the occurrence of the Peltzman effect during the COVID-19 outbreak. Previous studies often emphasized that risk perception would influence reactions to COVID-19. For example, worry about being infected raised people’s willingness to receive vaccination [19], and people tend to underestimate the likelihood of being infected [29], which is consistent with an optimism bias [34]. However, these studies did not look into the relationship between protective measures (e.g., whether there is a cooperative or a competitive relation). In this study, we found that receiving vaccination might lower people’s alertness, which, in turn, reduces their intention to adopt personal NPIs, which supports the existence of the Peltzman effect.

In addition to our study, a preprint paper also discussed the Peltzman effect on personal NPIs induced by vaccination [35]. However, our approach differs, as we applied a structural model to account for how the Peltzman effect occurs. As shown in Figure 2, Vaccine is not correlated with Worry in Survey 1, whereas they are negatively correlated in Survey 2. This correlation shift implies that people’s attitudes toward vaccines influence their worry level only after they understand how vaccination can protect them during a COVID wave: the more positive one’s attitude toward vaccines, the less worried one is. As Worry positively predicts NPIs in each sample, the less worried one is, the less likely one is to adopt personal NPIs. This mechanism elucidates how the Peltzman effect occurred in our study, which is mediated by the emotional component of risk perception.

At last, our results explain why about half of Taiwanese participants expressed high motivation to receive vaccination but low motivation to adopt personal NPIs [12]. Despite general acceptance of mask-wearing in public, individuals might deem it unnecessary once vaccinated, reflecting the Peltzman effect. Our findings may aid the Taiwanese government in enhancing epidemic prevention communication. Emphasizing factors like immune evasion or the fast mutation of SARS-CoV-2 could increase people’s sense of crisis, potentially increasing their intention to implement both preventive measures.

### Study Strengths and Limitations

Several strengths of this study are noteworthy. First, Taiwan provides an ideal setting for testing the Peltzman effect, given its stringent prevention policy for COVID-19. The population had received sufficient vaccinations before the COVID-19 outbreak, allowing for a natural-experiment-like study. This design minimizes extraneous variables and confounding factors. Second, this study applied the theory of risk perception and the risk-as-feelings model to explain and account for the observed Peltzman effect. This approach enhances the theoretical framework and provides a nuanced understanding of the psychological mechanisms involved in the Peltzman effect. Third, we quantified the Peltzman effect via the structural model as the indirect influence of people’s attitudes toward COVID-19 vaccines on their intention to adopt personal NPIs. 

This study has a few limitations that should be considered when interpreting the results. Firstly, the cross-sectional, time-sequential survey design used in the current study may introduce potential sampling bias and alternative explanations for the observed differences between the two surveys, as two different groups of participants were recruited in two data collections. However, a few factors indicated that these two datasets were comparable. First, the multigroup SEM results indicated that both survey groups shared the same mental constructs and processing routes. Moreover, there was no age difference between the two groups. In addition, we distributed questionnaires on the same platforms at two time points, thus the same user groups were accessed. Therefore, we believe that the current cross-sectional approach can still bring some useful insights into the dynamics of risk perception during the pandemic, just like previous studies employing cross-sectional and time-sequential surveys to study a cohort’s attitudes toward and reactions to COVID-19 [19,36,37,38].

Secondly, the use of single items instead of developed questionnaires for each mental construct can be seen as a limitation. Each of the mental constructs was assessed using three to four questions. Although normally using single items has some disadvantages, such as reduced precision and reliability, it is worth noting that the composite reliabilities and the construct validities were satisfactory in both surveys. Moreover, using single items allows for quicker data collection and minimizes the burden on participants, which is crucial given the rapidly changing pandemic situations. Similar usage of single items has been also seen in previous research [17,18,19,39], and we believe the same reasons apply to these studies. In addition, as a majority of participants were recruited from social media platforms, our samples are biased toward younger adults between the ages from 18 to 40. Thus, caution should be exercised when applying the current findings to older populations.

Lastly, the current study focused on specific aspects of risk perception, attitudes toward vaccines, and intention to adopt personal NPIs in response to the evolving pandemic. Other factors, such as cultural, political, or socioeconomic influences, were not thoroughly examined in this research. These unmeasured factors could potentially contribute to the observed behaviors and attitudes, and further research is needed to explore their impact on risk perception and preventive behaviors during the pandemic.

## 5. Conclusions

In this study, we verified the hypothesis that the decline in the adoption of personal NPIs is indeed attributed to the Peltzman effect in relation to vaccination. That is, having received COVID-19 vaccinations decreases people’s worry about COVID-19, which, in turn, decreases the intention to adopt personal NPIs, as they might think that they have had enough protection through vaccination. As a result, they might think it less necessary to adopt personal NPIs. Different from previous studies, our results highlighted a possible competitive relationship between preventive measures and could aid governments in enhancing epidemic prevention communication.

## Figures and Tables

**Figure 1 medicina-60-00301-f001:**
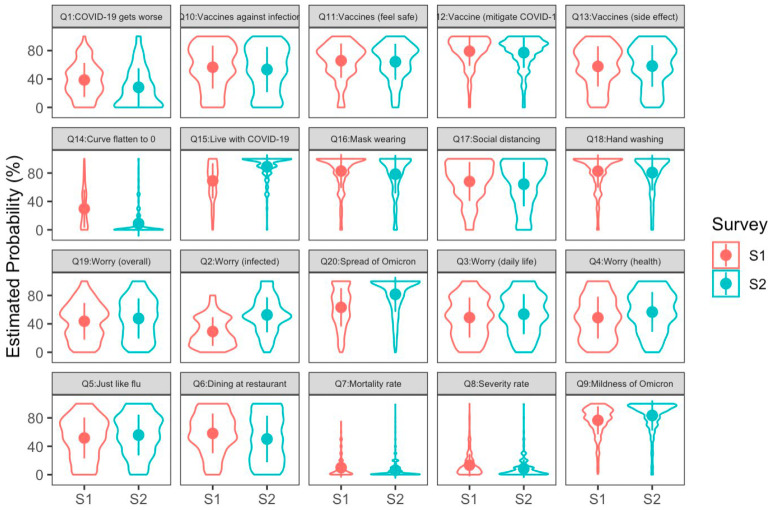
Violin plots for the distributions of participants’ ratings for all items in two surveys. The dot denotes the mean, and the error bar is one SD of the mean.

**Figure 2 medicina-60-00301-f002:**
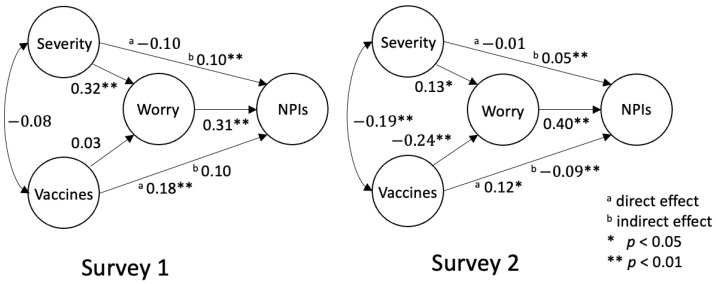
Structure models for data in two surveys. Severity: Perceived severity of COVID-19. Worry: Worry about COVID-19. Vaccines: Attitudes toward COVID-19 vaccines. NPIs: Intention to adopt personal NPIs. All weights are standardized scores with the latent variables and indicators all being normalized.

**Table 1 medicina-60-00301-t001:** Mean responses to items corresponding to age, worry, perceived severity, NPI adoption, vaccine attitudes, and number of doses of COVID-19 vaccines received.

Item	Survey 1	Survey 2	t	z	p Value
Age (years)	28.67	28.41	0.51		=0.610
Worry Infected (Q2) Daily life (Q3) Health (Q4) Overall (Q19) Cronbach’s α	29.19 48.89 48.71 43.65 0.79	52.64 53.63 56.68 47.60 0.78	−16.05 −2.61 −4.32 −2.27		<0.001 ** =0.009 * <0.001 ** =0.024 *
Severity Mortality (Q7) Fatality (Q8) Mildness (Q9) Cronbach’s α	9.91 13.45 76.61 0.74	6.14 8.49 83.44 0.66	4.56 5.36 −5.17		<0.001 ** <0.001 ** <0.001 **
Intention to adopt NPIs Mask-wearing (Q16) Social distancing (Q17) Handwashing (Q18) Cronbach’s α	82.77 68.63 82.63 0.80	78.41 64.32 80.64 0.82	2.64 1.99 1.27		=0.008 ** =0.047 * =0.205
Vaccine Effective for infection (Q10) Safety (Q11) Effective for severity (Q12) Cronbach’s α	56.63 65.80 79.15 0.78	53.36 64.30 77.15 0.75	1.63 0.94 1.47		=0.103 =0.349 =0.142
Dose received 0 1 2 3	4.76 3.90 67.75 23.59	2.97 2.18 15.64 79.20		1.45 1.57 16.49 −17.30	=0.147 =0.117 <0.001 ** <0.001 **

* *p* < 0.05, ** *p* < 0.01.

**Table 2 medicina-60-00301-t002:** Measurement model for both samples with the marker method for parameter estimation. The loadings are estimated using the marker method.

Latent Variables	Indicator Items	Loadings	p
Worry	Worry about getting infected (Q2)	1.00	
	Worry about daily life being influenced (Q3)	2.93	<0.01
	Worry about health being negatively affected (Q4)	2.78	<0.01
	Overall worry about COVID-19 (Q19)	2.69	<0.01
Severity	COVID fatality rate (Q7)	1.00	
	COVID severity rate (Q8)	1.09	<0.01
	COVID mildness (Q9)	−0.47	<0.01
Vaccines	Vaccination can decrease COVID infection (Q10)	1.00	
	Feeling safe with COVID vaccines (Q11)	1.67	<0.01
	Vaccination can decrease COVID severity rate (Q12)	1.17	<0.01
NPIs	Masks-wearing (Q16)	1.00	
	Social distancing (Q17)	1.05	<0.01
	Handwashing (Q18)	0.85	<0.01

## Data Availability

The data of this study can be found at https://osf.io/47khw/?view_only=17d668b595bf49bcab2fdd83fb54d878 (accessed on 30 December 2023).

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
