# Peer review of "Why Do We Not Wear Masks Anymore during the COVID-19 Wave? Vaccination Precludes the Adoption of Personal Non-Pharmaceutical Interventions: A Quantitative Study of Taiwanese Residents"

_medicina, 2024, doi:10.3390/medicina60020301_

Round 1

Reviewer 1 Report

Comments and Suggestions for Authors

Thank you for the opportunity to review the manuscript. The topic of nonpharmaceutical interventions (NPIs) is interesting to explore.

Abstract:

·        Generally, there is no citation in the abstract; therefore, I suggest not citing in the abstract.

·        Please present results with z and p values in the abstract.

Introduction:

·        Major Concern: The introduction section lacks a literature review, the context of writing this paper in the Taiwanese context, the rationale for conducting research, and a theoretical overview. The author must structure the paper based on these criteria. The current introduction of this paper is inadequate.

·        Some minor concern: The statement "However, people seemed to be less favorable to NPIs" needs clarification on who the people are. Please rewrite it for more clarity. Also, include a citation for "However, a small proportion of risk-seeking individuals cannot fully explain the widespread resistance to personal NPIs (e.g., mask-wearing)."

Materials and Methods:

·        Regarding the study sample (Survey 1), please explain who collected the survey, how the validity and reliability of the survey questionnaire were maintained.

·        Explain the representativeness of the survey and how data collection was conducted. Clarify how variables were coded for better understanding in the Results section.

Results:  

·        Move Table 1 to the Appendix. In the Results section, the use of quantitative research analyses confuses with the topic mentioning "A systematic view based on risk perception." Explain the concept of "systematic view" in the introduction section for clearly understanding and interpreting the results.

Discussion:  

·        In the Discussion section, elaborate on the impact of the study on non-pharmaceutical interventions, particularly in the Taiwanese context.

·        Strengthen the discussion on non-pharmaceutical interventions.

·        In the Study Limitations section, add strengths and discuss how the findings contribute to the current literature and their potential applications in social or health policy development.

Conclusion:

·        Rewrite the conclusions to provide readers with the significance of the research's impact on the field, emphasizing key takeaways, implications, and potential avenues for future research.

Author Response

We appreciate your time and efforts for generating these valuable comments. The point-by-point responses can be seen in the WORD file we uploaded and our revised manuscript. As we would like to highlight the relationship between NPIs and vaccination, we changed the title of our manuscript too.

Reviewer 2 Report

Comments and Suggestions for Authors

I have carefully reviewed the manuscript, titled “Factors affecting the decline in personal non-pharmaceutical intervention adoption during the COVID-19 wave: A systematic view 3 based on risk perception”. The study was to examine an explanation for factors affecting the decline in personal non-pharmaceutical intervention adoption during the COVID-19. It was based on the framework of risk perception.

Although the study has some strong points, some points need to be addressed.

Introduction:

1) The Introduction section should be more detailed and precise, with more information on factors affecting the decline in personal non-pharmaceutical intervention.

2) Can you present psychological and medical mechanisms responsible nonpharmaceutical interventions? What are its underlying mechanisms (pp. 1-2).

3) It would be beneficial to provide more information on the association nonpharmaceutical interventions with risk perception.

4) Hypotheses are properly formulated to specify the aim of the study.

Method:

5) Was the sample determined by power analysis?

6) There should be psychometric values for the questionnaire used in your study.

7) How did you handle missing values in your data? (If any exist)

Results:

8) The results are properly presented.

Discussion:

9) What are the underlying mechanisms responsible for this result: “we demonstrated through SEM the path how vaccination induces the laxity in personal NPIs via lowering people’s worry about COVID-19.”?

10) The results obtained in the study should be more thoroughly discussed in the context of risk perception (p. 9).

 11) This statement is rather startling: “Is it possible to have a reversed Peltzman effect on receiving vaccination?” (p. 9). How would you explain it?

Author Response

(The authors gave the same response as above.)

Round 2

Reviewer 1 Report

Comments and Suggestions for Authors

Now, the manuscript is much improved, and it can be acceptable.